# Discontinuation of oral anticoagulation therapy after successful atrial fibrillation ablation: A systematic review and meta-analysis of prospective studies

Xue-Hui Liu[☯], Qiang Xu[☯], Tao Luo, Lei Zhang, Hong-Jun Liu[ID]*

Department of Cardiology, Traditional Chinese Medicine Hospital of China Three Gorges University, Yichang Hospital of Traditional Chinese Medicine, Yichang, China

☯ These authors contributed equally to this work.
* yczyylhj@sina.com

**Data Availability Statement:** All relevant data are within the paper and its S1 Checklist, S1 Fig, S1–S3 Tables.

## Abstract

### Background

The safety of discontinuing oral anticoagulant (OAC) therapy after atrial fibrillation (AF) ablation remains controversial. A meta-analysis was performed to assess the safety and feasibility of discontinuing OAC therapy after successful AF ablation.

### Methods

PubMed and Embase were searched up to October 2020 for prospective cohort studies that reported the risk of thromboembolism (TE) after successful AF ablation in off-OAC and on-OAC groups. The primary outcome was the incidence of TE events. The Mantel-Haenszel method with random-effects modeling was used to calculate pooled odds ratios (ORs) and 95% confidence intervals (CIs).

### Results

A total of 11,148 patients (7,160 in the off-OAC group and 3,988 in the on-OAC group) from 10 studies were included to meta-analysis. No significant difference in TE between both groups was observed (OR, 0.73; 95%CI, 0.51–1.05; $I^2 = 0.0\%$). The risk of major bleeding in off-OAC group was significantly lower compared to the on-OAC group (OR, 0.18; 95%CI, 0.07–0.51; $I^2 = 51.7\%$).

### Conclusions

Our study suggests that it may be safe to discontinue OAC therapy in patients after successful AF ablation. Additionally, an increased risk of major bleeding was observed in patients on OAC. However, the results of this meta-analysis should be interpreted with caution because of the heterogeneity among the included study designs. Large-scale and adequately powered randomized controlled trials are warranted to confirm these findings.

**Funding:** The author(s) received no specific funding for this work.

**Competing interests:** The authors have declared that no competing interests exist.

**Abbreviations:** AF, atrial fibrillation; OAC, oral anticoagulant; DOAC, direct oral anticoagulant; TE, thromboembolism; OR, odds ratio; CI, confidence interval; NOS, Newcastle-Ottawa Scale.

# 1. Introduction

Atrial fibrillation (AF) is the most common cardiac arrhythmia, which is associated to increased morbidity and mortality [1–3]. Drug therapy is considered a first-line strategy for the management of AF [2]. However, the increasing number of AF patients and recognition of increased morbidity, mortality, impaired quality of life, and side effects of antiarrhythmic drugs have spurred numerous investigations to develop more effective treatments for AF and its complications [1]. Catheter ablation is increasingly being used for rhythm management in AF patients. It is regarded as an effective intervention for improving patients' clinical symptoms, reducing the AF burden and cardiovascular hospitalizations, and improving patients' quality of life [4]. Despite high rates of sinus rhythm maintenance, the optimal anticoagulation therapy strategy after AF ablation is still undetermined.

To date, a large number of published observational studies [5–8] have supported the discontinuation of oral anticoagulation (OAC) in patients after successful AF ablation. According to current guidelines [2], OAC therapy is recommended for at least two months post ablation in all patients. Beyond this time period, a decision to continue OAC is determined by the presence of congestive heart failure, hypertension, age> 75 years, diabetes mellitus, stroke or transient ischemic attack (TIA), vascular disease, age 65 to 74 years, sex category ($CHA_2DS_2$-$VAS_c$) score rather than the rhythm status [2]. However, the safety of this strategy has not been proven in large randomized trials. To evaluate the safety and effectiveness of discontinuing of OAC after successful AF ablation, we performed the present study by systematic review and meta-analysis of prospective studies.

# 2. Methods

## 2.1. Literature search

Pubmed and Embase (from inception to October 2020) were searched to identify studies comparing the discontinuation vs. use of OAC therapy after successful AF ablation. No language restriction was applied. Searched terms included "atrial fibrillation," "ablation," "anticoagulation," "anticoagulant," "Warfarin," "dabigatran," "apixaban," "rivaroxaban," and "edoxaban". The detailed search strategy is presented in S1 Table. In addition, the reference lists of identified articles were manually screened for potential studies. Our systematic review and meta-analysis were conducted according to the checklist of Preferred Reporting Items for Systematic Reviews and Meta-Analysis (PRISMA) statement [9].

## 2.2. Study selection

Studies were considered acceptable if they met the following criteria: (1) prospective cohort study; (2) reported the effects of off-OAC and on-OAC in patients after successful AF catheter ablation; (3) primary outcome: incidence of TE events (including stroke and TIA); and (4) secondary outcome: incidence of major bleeding. Articles were excluded if they (1) were reviews, abstracts, letters, or conference abstracts; and/or (2) did not report outcomes of interest.

## 2.3. Data extraction and quality assessment

Two authors independently extracted data from all included studies using a standardized Excel file. Information was extracted on: (1) first author, baseline characteristics of participants, year, duration of follow-up, geographical location, type of AF, catheter ablation and OAC strategy, definition of AF recurrence; (2) primary and secondary outcome; (3) confounding variables, $CHADS_2$ and/or $CHA_2DS_2$-$VAS_c$ score.

The quality of each study was evaluated using the Newcastle-Ottawa Scale (NOS) [10]. The quality score of studies was calculated based on three components: selection of the study groups (0–4 points), comparability of study groups (0–2 points), and ascertainment of the outcome of interest (0–3 points). The score ranges from 0 to 9 points. A higher score indicated better methodological quality. Disagreements were resolved by discussion.

## 2.4. Statistical analysis

As the incidence of TE events and major bleeding were rare, odds ratios (ORs) could be assumed to be accurate estimates of risk ratios. The Mantel-Haenszel method with random-effects modeling was used to calculate pooled ORs and 95% confidence intervals (CIs). Heterogeneity among the studies was assessed using the $I^2$ statistic [11], where $I^2$ values of 25%, 50%, and 75% corresponded to cut-off points for slight, moderate and high degree of heterogeneity. When evident heterogeneity was present, we assessed the influence of a single study on the overall pooled effect by omitting one study in each turn. Subgroup analysis was performed to test the robustness of pooled effects. The publication bias was assessed by using both Begg's test and Egger's test. A two-tailed $P$-value $< 0.05$ indicated statistical significance. All statistical analyses were performed using Stata 12.0 (StataCorp, College Station, TX, USA).

# 3. Results

## 3.1. Search results

Our literature search identified 2504 articles. After a review of titles and abstracts, 2,472 studies were excluded, and the remaining 32 were considered potentially eligible trials and identified by reading the full-text. Finally, a total of 10 studies [12–21] were included. After screening the reference lists of included articles, we retrieved two potential studies, but neither met our inclusion criteria. Overall, 10 studies enrolling 11,148 participants were included in the meta-analysis. Fig 1 shows the detailed search strategy.

## 3.2. Characteristics of included studies

The characteristics of included studies are summarized in Table 1. These studies were published between 2006 and 2020. Among the ten studies included here, four were conducted in the United States [12–14, 16], two in China [19, 20], one in the UK and Australia [21], one in France [18], and one in Italy [17]. The sample size ranged from 108 [16] to 4512 [19] patients. The majority of the participants were male. The proportion of men ranged from 61.6% [20] to 79.9% [17]. Only one [20] study enrolled patients with paroxysmal AF. The duration of follow-up across the studies ranged from 1.9 to 5 years. Warfarin as the only anticoagulant drug was prescribed for patients in all but four studies [16, 18–20], which included partial patients on direct OACs (DOACs). OAC was discontinued in 7,160 (64.2%) patients. Eight studies [12–18, 21] reported the blanking period, which ranged from two to three months. The time frame of discontinuation of OAC ranged from 2 to 12 months. In nine studies collectively [12–16, 18–21], 2488 patients (24.0%) developed AF recurrence. In six studies [13, 15–17, 19, 21], OAC was switched to antiplatelet agents in the majority of off-OAC patients. Two studies [17, 19] stratified patients according to the $CHA_2DS_2-VAS_c$ score and reported corresponding clinical outcomes in the two groups. Nine studies [12–20] were assessed to be high quality according to the NOS score (range 7 to 9). The results of the quality assessment are described in S2 Table.

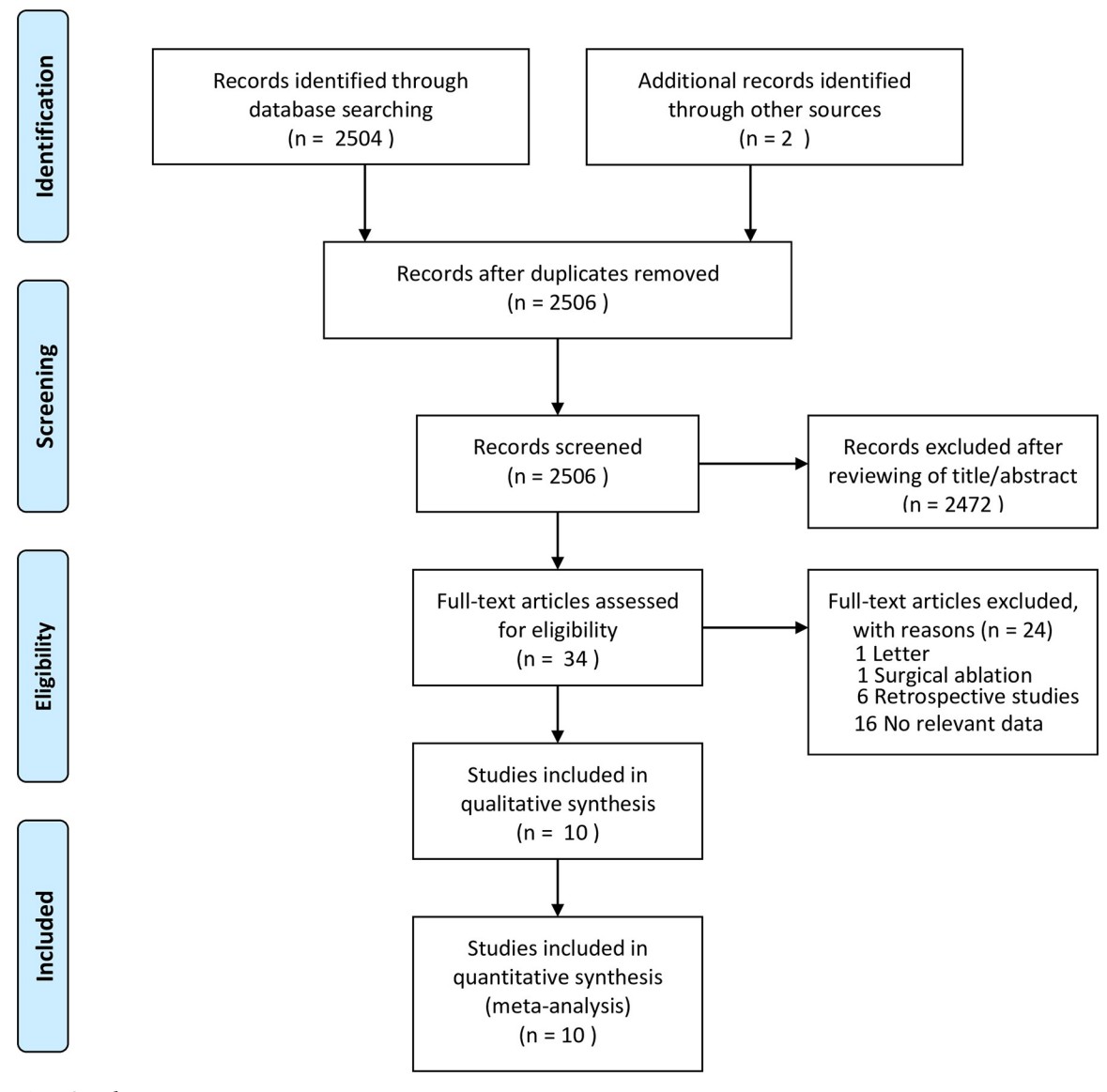

**Fig 1. Search strategy.**

### 3.3. Risk of TE events in off-OAC vs on-OAC patients after AF ablation

A total of 10 studies provided this outcome [12–21]. In the off-OAC group, 77 patients (1.1%) suffered from TE events, while the on-OAC group had 56 patients (1.4%) who suffered from TE events. The pooled OR for TE events in patients after AF ablation was 0.73 (95% CI: 0.51–1.05) (Fig 2). There was no heterogeneity between the studies ($I^2$ = 0.0%, P = 0.478). No significant publication bias was found according to Begg's and Egger's tests (both P values > 0.05).

To test the robustness of the meta-analysis, subgroup analysis was performed according to the study region (non-Asian vs. Asian), sample size (<800 vs. >800), mean age ($\leq$ 60 years vs. >60 years), OAC strategy (warfarin vs. warfarin or DOAC) and follow-up duration (<3 years vs. >3years) (Table 2).

**Table 1. Characteristics for studies included in the meta-analysis.**

| Author and year | Region | Type of OAC | Type of ablation (energy) | Blanking Period (months) | Sample size | Age, years (range) | Male N (%) | Follow-up (years) | Time of off-OAC (months) | Reinitialization of OAC after AF recurrence | CHADS$_2$ score off-OAC | CHADS$_2$ score on-OAC |
|---|---|---|---|---|---|---|---|---|---|---|---|---|
| Oral et al., 2006 [12] | USA | Warfarin | CA (RF) | 2 | 755 | 55±11 (17–79) | 577 (76.4%) | 2.1±0.7 | 3 | Unclear | 0 = 53% ≥1 = 47% | 0 = 37% ≥1 = 63% |
| Nademanee et al., 2008 [13] | USA | Warfarin | CA (RF) | 3 | 635 | 67±12 | 423 (66.5%) | 2.3+1.7 | 3 | Restarted | NA | NA |
| Hussein et al., 2011 [14] | USA | Warfarin | CA (RF) | 2 | 831 | 58.7±9.9 | 644 (77.5%) | 4.6* | 12 | Unclear | NA | NA |
| Hunter et al., 2011 [21] | UK/Australia | Warfarin | CA (RF+Cryo) | 3 | 1273 | 58±11 | 942 (74%) | 3.1 | 3 | unclear | 0.7±0.9 | 0.9±0.9 |
| Saad et al., 2011 [15] | Brasil | Warfarin | CA (RF) | 2 | 327 | 63±13 (17–87) | 259 (79.2%) | 3.8±1.4 | 3 | Restarted | NA | NA |
| Winkle et al., 2013 [16] | USA | Warfarin or DOAC | CA (RF) | 3 | 108 | 66.2±9.0 | 68 (62.9%) | 2.8+1.6 | 7 | Unclear | NA | NA |
| Gaita et al., 2014 [17] | Italy | Warfarin | CA (RF) | 3 | 766 | 57±11 | 612 (79.9%) | 5* | 3 | Restarted | ≤1 = 91.8% ≥2 = 8.2% | ≤1 = 70.4% ≥2 = 29.6% |
| Hermida et al., 2020 [18] | France | Warfarin or DOAC | CA (Cyro) | 3 | 450 | 60±9 | 351 (78%) | 2.2* | 3 | Unclear | 0.7±1.0# | 1.8±1.3# |
| Yang et al., 2020 [19] | China | Warfarin or DOAC | CA (RF) | NA | 4512 | On-OAC 64.1±9.8 Off-OAC 62.8±9.9 | 2864 (63.5%) | Off-OAC 2.0±1.2 On-OAC 1.9±1.1 | 12 | Restarted | 2.3±1.3# | 2.7±1.4# |
| Yu et al., 2020 [20] | China | Warfarin or DOAC | CA (RF) | NA | 1491 | 59.6±12.1 (43–76) | 918 (61.6%) | 2.3±1.2 | 3 | Physician discretion | 1.5±1.4# | 2.4±1.7# |

CA, catheter ablation; Cryo, cryo-balloon; NA, data not available; DOAC, direct oral anticoagulation; OAC, oral anticoagulant; RF, radiofrequency

#, CHADS$_2$-VASc score

*, median.

### 3.4. Risk of major bleeding in off-OAC vs. on-OAC patients after AF ablation

Seven studies [12, 13, 15–17, 19, 20] that enrolled 8,594 participants were included in the meta-analysis of major bleeding. During the follow-up, major bleeding events occurred in 33 out of the 5,827 off-OAC patients (0.6%) and in 44 out of the 2,767 on-OAC patients (1.6%). The pooled OR for major bleeding was 0.18 (95% CI: 0.07–0.51) (Fig 3). There was moderate heterogeneity between the studies ($I^2$ = 51.7%, P = 0.053). Exclusion of any single study did not materially alter the overall combined OR. Due to the number of included studies < 10, the publication bias was not performed.

## 4. Discussion

There is a general consensus that OAC therapy should be continued for least two months in all patients after AF ablation [1, 2]. Beyond this time, the use of OAC is controversial. Current guidelines recommend that a decision for long-term OAC therapy is determined by the presence of CHA$_2$DS$_2$-VAS$_c$ stroke risk factors rather than the rhythm status [2].

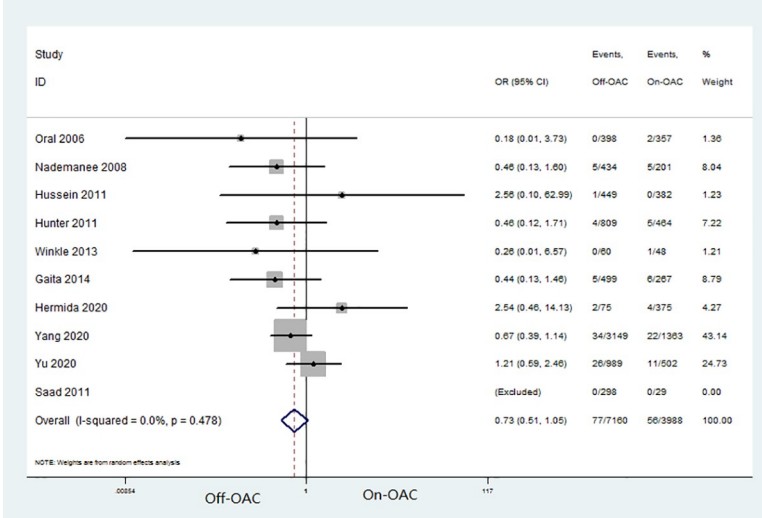

**Fig 2. Forest plot for TE event in off-OAC vs. on-OAC patients after AF ablation.** TE, hromboembolism; OAC, oral anticoagulant; AF, atrial fibrillation.

Our study assessed the safety of discontinuing OAC in patients after successful AF ablation by meta-analysis of prospective studies. There is no significant difference in the incidence of TE events between both groups. However, the risk of major bleeding was seen to be higher in the on-OAC group, with an OR reduction of 82% in the off-OAC group. These findings are similar to previous meta-analyses [22, 23]. After the publication of these studies, however, several large-scale prospective studies were published and involved more confident evidence. Compared with previous studies, we only selected prospective studies and included the latest three studies [18–20] (including the largest scale study on this topic), which further reinforces earlier outcomes.

The incidence rates of TE events in off-OAC and on-OAC group were 1.1% and 1.4%. These results are similar to the general population (1.4%) [24]. Bunch et al. reported that AF patients with ablation had a lower risk of stroke compared to those without ablation (1.4% vs. 3.5%) [24]. Our results suggest that it may be safe to discontinue OAC in patients, including those with a high risk of TE, following successful AF ablation. However, due to lack of patient-

**Table 2. Subgroup analyses based on various variables for the safety of off-OAC after AF ablation.**

| Subgroup | Categorical data | NO. studies | OR (95%CI) | Heterogeneity test ($I^2$, P) |
|---|---|---|---|---|
| Ethnicity | Non-Asian [12–18, 21] | 8 | 0.57 (0.31, 1.07) | 0.0%, P = 0.549 |
| | Asian [19, 20] | 2 | 0.85 (0.48, 1.52) | 41.1%, P = 0.193 |
| Sample size | <800 [12, 13, 15–18] | 6 | 0.57 (0.27, 1.18) | 0.6%, P = 0.403 |
| | >800 [14, 19–21] | 4 | 0.80 (0.53, 1.19) | 0.0%, P = 0.404 |
| Mean age | ≤60 years [12, 14, 17, 18, 20, 21] | 6 | 0.88 (0.47, 1.58) | 13.9%, P = 0.326 |
| | >60 years [13, 15, 16, 19] | 4 | 0.62 (0.38, 1.01) | 0.0%, P = 0.761 |
| OAC strategy | Warfarin [12–15, 17, 21] | 6 | 0.47 (0.23, 0.93) | 0.0%, P = 0.830 |
| | Warfarin or DOAC [16, 18–20] | 4 | 0.91 (0.54, 1.54) | 20.7%, P = 0.286 |
| Duration of follow-up | <3 years [12, 13, 16, 18–20] | 6 | 0.80 (0.51, 1.25) | 11.1%, P = 0.344 |
| | >3 years [14, 15, 17, 21] | 4 | 0.51 (0.22, 1.19) | 0.0%, P = 0.587 |

DOAC, direct oral anticoagulant; OAC, oral anticoagulant; OR, odds ratio; CI, confidence interval.

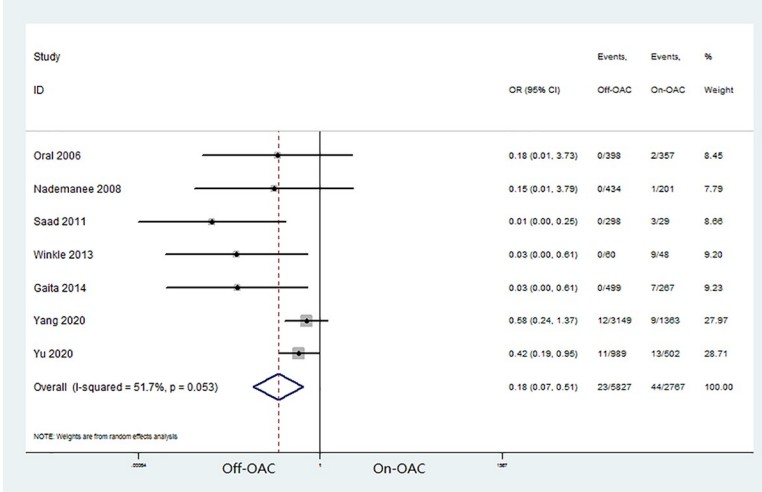

**Fig 3. Forest plot for major bleeding in off-OAC vs. on-OAC patients after AF ablation.** OAC, oral anticoagulant; AF, atrial fibrillation.

level data, the non-significant incidence of TE in the off- vs. on-OAC groups needs to be interpreted cautiously. Potential mechanisms of TE reduction after AF ablation may be related to the rhythm control and favorable remodeling of the atrium [2]. Rates of AF progression were significantly lower with rhythm control than that with rate control [25]. Thus, a successful ablation may offer an opportunity to halt the progressive patho-anatomical changes and reduce the risk of TE [26]. The Early Treatment of Atrial Fibrillation for Stroke Prevention Trial (EAST-AFNET 4) demonstrated that early rhythm-control therapy was associated with a lower risk of death from cardiovascular causes and stroke than usual care over a follow-up time of more than five years [27].

Of all included studies, two [13, 16] evaluated the long-term clinical outcomes of AF ablation in high-risk patients. The pooled analysis showed that off-OAC was not associated with an increased risk of TE in high-risk patients (OR: 0.43, 95% CI: 0.13–1.38). A similar result has been reported by Proietti et al. in a meta-analysis, which showed no significant clinical benefit of on-OAC in patients with $CHADS_2/CHA_2DS2 – VAS_c \geq 2$ [23]. In addition, the China-AF study is one of the largest prospective registry studies (including 4,512 patients) of Asian patients with AF [19], and reported that TE rates after AF ablation in the off-OAC group were not significantly different from the on-OAC group among high-risk patients (female, $CHA_2DS2 – VAS_c \geq 3$; male, $CHA_2DS2 – VAS_c \geq 2$). Conversely, a meta-analysis by Romero et al. reported that on-OAC after AF ablation with $CHA_2DS_2 -VAS_c$ score $\geq 2$ is associated with a significantly decreased risk of TE and a favorable net clinical benefit [28]. This discrepancy may be related to follow-up duration and heterogeneity among study designs.

The subgroup analysis by type of OAC showed that the discontinuation of warfarin would reduce the risk of TE compared with long-term treatment in patients after AF ablation. One possible reason may be the higher cardiovascular risk profile of the patients in the on-OAC group as compared with that of the off-OAC group (e.g. patients in the on-OAC group were older [12, 14, 15, 17–20], with higher $CHADS_2/CHA_2DS_2 -VAS_c$ scores [12, 14, 17–21] and with a prevalence of persistent or longstanding persistent AF [13, 15, 17–19] compared with those of the off-OAC group). History of older age, heart failure, hypertension, diabetes, and stroke were independently associated with an increased risk of TE; hence, high-risk patients still remained a significant risk factor after successful AF ablation [2, 28].

All but one of the included studies had consisted predominantly of men. As we know, sex-related differences in the epidemiology, pathophysiology, clinical presentation, and prognosis of AF may influence the effectiveness of AF therapy [2]. It has been reported that women undergoing AF ablation were older with more comorbidities and less paroxysmal AF [29], which could lead to lower success rates. Moreover, women experienced higher AF recurrence than that in men after successful ablation [30]. A meta-analysis [31] of 14 studies demonstrated that women were associated with an increased risk of stroke/TIA compared with that in men (0.51% vs. 0.41%, OR: 1.42, 95% CI: 1.21–1.67, P < 0.0001). In addition, the sex difference in the risk of stroke/TIA was independent of follow-up time [31]. Owing to the under-representation of women, our findings should be interpreted cautiously regarding female patients.

Long-term OAC therapy can result in severe bleeding complications, and the decision on whether it is safe to discontinue OAC after successful ablation remains controversial. In the present study, a significant increase in episodes of major bleeding was observed in the on-OAC group compared with that in the off-OAC group. Therefore, it is important to balance the TE and the major bleeding risk in patients after AF ablation, particularly when coupled with a high $CHA_2DS_2\text{-}VAS_c$ score. Of all of the included studies, warfarin was the only anticoagulant in all but four studies [16, 18–20], which included partial patients on DOACs. DOACs are relatively new drugs demonstrating noninferiority or superiority to warfarin in reducing risk TE events with a similar or reduced bleeding risk [2, 32]. It may imply a greater net clinical benefit derived from long-term DOACs therapy in the majority of those with a high TE risk. The results of the ongoing Optimal Anticoagulation for Higher-Risk Patients Post-Catheter Ablation for Atrial Fibrillation (OCEAN) study [33] will hopefully provide concrete evidence for this topic.

## 5. Limitations

There are several limitations in our meta-analysis that should be acknowledged. First, because continuous monitoring systems were not available for all of the patients, some asymptomatic AF recurrences may have been unrecognized. It may lead to potential underestimation of the incidence of AF recurrence and overestimation of risk of long-term on-OAC therapy. In addition, some patients may have died before receiving hospital care, and the underlying cause may have gone undetermined, leading to an underestimation of the incidence rates. Second, the risks of TE and major bleeding in patients with different $CHADS_2$ or $CHA_2DS2\text{ }\text{–}VAS_c$ scores were not evaluated according to the stratification of risk between both groups. Third, out of all of the included studies, six used warfarin as OAC therapy. Whether our findings can be extended to DOACs needs further investigations. Finally, there is the short follow-up duration of included studies, which could not represent the very long-term TE and major bleeding events.

## 6. Conclusions

The present study suggests that it may be safe to discontinue OAC in patients after successful AF ablation. Additionally, an increased risk of major bleeding was observed in on-OAC patients. However, the results of this meta-analysis should be interpreted with caution because of the heterogeneity among the included study designs. Our results support that large-scale randomized controlled trials are warranted to confirm these findings.

## Supporting information

**S1 Checklist. PRISMA 2009 checklist.**
(DOC)

**S1 Fig. Publication bias for the thromboembolic events.**
(PDF)

**S1 Table. Literature search strategy.**
(PDF)

**S2 Table. The Newcastle-Ottawa scale of individual study.**
(PDF)

**S3 Table. Incidence of thromboembolism and major bleeding in the included studies.**
(PDF)

## Author Contributions

**Conceptualization:** Xue-Hui Liu, Qiang Xu, Tao Luo, Hong-Jun Liu.

**Data curation:** Xue-Hui Liu, Qiang Xu.

**Formal analysis:** Xue-Hui Liu, Lei Zhang.

**Methodology:** Xue-Hui Liu, Qiang Xu, Lei Zhang.

**Supervision:** Hong-Jun Liu.

**Writing – original draft:** Xue-Hui Liu.

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
