## [Decision Letter · Decision Letter 0]

1 Apr 2021

PONE-D-21-00884

Discontinuation of oral anticoagulation therapy after successful atrial fibrillation ablation: a systematic review and meta-analysis of prospective studies

PLOS ONE

Dear Dr. Liu,

Thank you for submitting your manuscript to PLOS ONE. After careful consideration, we feel that it has merit but does not fully meet PLOS ONE’s publication criteria as it currently stands. Therefore, we invite you to submit a revised version of the manuscript that addresses the points raised during the review process.

We look forward to receiving your revised manuscript.

Kind regards,

Yoshihiro Fukumoto

Academic Editor

PLOS ONE

Journal Requirements:

PLOS requires an ORCID iD for the corresponding author in Editorial Manager on papers submitted after December 6th, 2016. Please ensure that you have an ORCID iD and that it is validated in Editorial Manager. To do this, go to ‘Update my Information’ (in the upper left-hand corner of the main menu), and click on the Fetch/Validate link next to the ORCID field. This will take you to the ORCID site and allow you to create a new iD or authenticate a pre-existing iD in Editorial Manager. Please see the following video for instructions on linking an ORCID iD to your Editorial Manager account: https://www.youtube.com/watch?v=_xcclfuvtxQ

We noticed you have some minor occurrence of overlapping text with the following previous publications, which needs to be addressed:

- https://academic.oup.com/europace/article/21/10/1509/5529272

- https://link.springer.com/article/10.1007/s11886-020-01313-1

- https://onlinelibrary.wiley.com/doi/abs/10.1111/jce.13822

- https://academic.oup.com/europace/article-abstract/22/1/90/5561458?redirectedFrom=fulltext

- https://academic.oup.com/eurheartj/article/42/5/373/5899003

In your revision ensure you cite all your sources (including your own works), and quote or rephrase any duplicated text outside the methods section. Further consideration is dependent on these concerns being addressed.

4. Within the manuscript text, please address the following:

4a) Please provide additional details on the inclusion and exclusion criteria for the included studies.

4b) Please provide the graphical results from the analysis of publication bias

4c) Please provide a table reporting in detail the results of your quality assessment, showing how each included study scored on every item of the Newcastle Ottawa Scale

4d) Please provide at least one detailed search/ Boolean strategy used in one of the database.

5, In your Data Availability statement, you have not specified where the minimal data set underlying the results described in your manuscript can be found. PLOS defines a study's minimal data set as the underlying data used to reach the conclusions drawn in the manuscript and any additional data required to replicate the reported study findings in their entirety. All PLOS journals require that the minimal data set be made fully available. For more information about our data policy, please see http://journals.plos.org/plosone/s/data-availability.

6. Please include a caption for figure 1.

7. Please include your tables as part of your main manuscript and remove the individual files. Please note that supplementary tables should be uploaded as separate "supporting information" files.

Reviewers' comments:

Reviewer's Responses to Questions

**Comments to the Author**

1. Is the manuscript technically sound, and do the data support the conclusions?

Reviewer #1: Yes

Reviewer #2: Yes

2. Has the statistical analysis been performed appropriately and rigorously? 

Reviewer #1: Yes

Reviewer #2: Yes

3. Have the authors made all data underlying the findings in their manuscript fully available?

Reviewer #1: Yes

Reviewer #2: Yes

4. Is the manuscript presented in an intelligible fashion and written in standard English?

Reviewer #1: Yes

Reviewer #2: Yes

5. Review Comments to the Author

Reviewer #1: General comment to the manuscript:

In the present study, Liu HJ et al. conducted a meta-analysis of the 10 studies testing wether discontinuation of OAC would be acceptable after successful AF ablation. The study results were that there was no significant increase of the thromboembolic events and that risk of major bleeding was significantly lower in the group of OAC discontinuation. The authors concluded that it may be safe to discontinue OAC in patients after successful AF ablation. A non-negligible limitation of the study is that most of the included studies are not with randomized nor case-control fashion with regard to the OAC treatment in the studied patients. Furthermore, similar study results have been reported previously (ref #24 and Romero J et al: JCE 2019 Aug;30(8):1250-1257.).

Major comments:

#1. In most of the included studies, OAC was discontinued only in patients with no AF recurrence after ablation and with low CHADS2/CHADS-Vasc score. The compared cohorts were with AF recurrence after ablation or with high thromboembolic scores, thus the comparisons were not matched with regard to clinical backgrounds but with considerable selection biases. The reviewer is questioned whether a meta-analysis containing such non-randomized heterogeneous studies leads to a reliable conclusion. The overall low incidence of thromboembolic events (1.1%) in the off-OAC patients after successful ablation is an important result to be known, however, a comparison to other cohort should be carefully done.

#2. Similar study results have been already reported previously (ref #24 and Romero J et al: JCE 2019 Aug;30(8):1250-1257.). The latter article should be cited in the present study. The authors should address what is a novelty and difference in comparison to these previous studies.

#3. The residual thromboembolic risk after AF ablation may vary according to the CHADS2/CHADS-Vasc score. Subgroup analyses with regard to these risk scores should be presented.

#4. A subgroup analysis with DOAC usage may be informative and give a novelty to the present study.

Minor comments:

#1. The discussion section is too long and should be simplified. The first para. is unnecessary because it was mentioned in the introduction.

#2. L217: Referring the paper by Dr. Bunch, stroke rate in patients who did not undergo ablation would be 3.5%.

#3. English usage should be revised through the manuscript. A native speaker check is recommended.

Reviewer #2: In this manuscript, authors showed that it may be safe to discontinue OAC (oral anticoagulation) therapy in patients after successful AF ablation. Additionally, an increased risk of major bleeding was observed in on-OAC patients.

Although this is interesting study, there are some problems as listed below.

As authors pointed out in “Limitation”, reviewer is afraid of that relatively short follow-up period and low CHADS2 score could affect the results of this study. Precise examination of thromboembolistic risks of patients after catheter ablation for atrial fibrillation (AF) would need data of longer follow-up period and one of relatively high-risk patients, for clinical settings. Moreover, the fact that OAC used in many studies was warfarin would also affect the results in this study. Because large number of patients who receive AF ablation are treated by DOACs in these days, data from warfarin usage could have some discrepancy with real world.

In clinical settings, various patients (CHADS2 score, comorbidity, age, and so on) exist, and complications from OAC usage or discontinuation, such as cerebral infarction and bleeding, are sometimes fatal. Therefore, reviewer thinks that discontinuation of OACs after AF ablation should be based on not crude or heterogeneous data, but detailed and precise one of each patient.

6. PLOS authors have the option to publish the peer review history of their article (what does this mean?). If published, this will include your full peer review and any attached files.

Reviewer #1: No

Reviewer #2: No

---

## [Author Response · Author response to Decision Letter 0]

30 Apr 2021

Response to journal requirements

Requirement: We noticed you have some minor occurrence of overlapping text with the following previous publications, which needs to be addressed:

-https://academic.oup.com/europace/article/21/10/1509/5529272

-https://link.springer.com/article/10.1007/s11886-020-01313-1

-https://onlinelibrary.wiley.com/doi/abs/10.1111/jce.13822

-https://academic.oup.com/europace/article-abstract/22/1/90/5561458?redirectedFrom=fulltext

-https://academic.oup.com/eurheartj/article/42/5/373/5899003

Response: Thank you for your reminder. We have checked, re-edited and quoted it in the revised manuscript.

Requirement: Within the manuscript text, please address the following:

4a) Please provide additional details on the inclusion and exclusion criteria for the included studies.

Response: We have added relevant data on the inclusion and exclusion criteria in the Figure 1.

4b) Please provide the graphical results from the analysis of publication bias

Response: Thank you for your advice. We have added the graphical results of publication bias in the supplementary materials.

4c) Please provide a table reporting in detail the results of your quality assessment, showing how each included study scored on every item of the Newcastle Ottawa Scale

Response: According your suggestion, the detailed NOS score have been provided in the supplementary materials (S2 Table).

4d) Please provide at least one detailed search/ Boolean strategy used in one of the database. 

Response: The literature search strategy was as follws:

#1 “atrial fibrillation” (title/abstract)

#2 “ablation” (title/abstract)

#3 “anticoagulation” (title/abstract)

#4 “anticoagulant” (title/abstract)

#5 (#3 OR #4)

#6 “dabigatran” (title/abstract)

#7 “apixaban” (title/abstract)

#8 “rivaroxaban” (title/abstract)

#9 “edoxaban” (title/abstract)

#10 (#6 OR #7 OR #8 OR #9)

#11 (#5 OR #10)

#12 (#1 AND #2 AND #11)

The search strategy is presented in the supplementary materials (S1 Table).

Requirement: In your Data Availability statement, you have not specified where the minimal data set underlying the results described in your manuscript can be found. PLOS defines a study's minimal data set as the underlying data used to reach the conclusions drawn in the manuscript and any additional data required to replicate the reported study findings in their entirety. All PLOS journals require that the minimal data set be made fully available. For more information about our data policy, please see http://journals.plos.org/plosone/s/data-availability.

Response: Our systematic review and meta-analysis was conducted according to the PRISMA statement. The completed PRISMA checklist and search flow diagram was provided in the supplementary materials and manuscript respectively. The characteristics of included studies were described in Table 1. Our study’s minimal data set was summarized in S3 Table. 

Requirement: Please include a caption for figure 1.

Response: We have added a caption for figure 1.

Requirement: Please include your tables as pacart of your main manuscript and remove the individual files. Please note that supplementary tables should be uploaded as separate "supporting information" files.

Response: We have added the tables in the revised manuscript.

Response to reviewers' comments

Reviewer #1

Comment: In most of the included studies, OAC was discontinued only in patients with no AF recurrence after ablation and with low CHADS2/CHADS-Vasc score. The compared cohorts were with AF recurrence after ablation or with high thromboembolic scores, thus the comparisons were not matched with regard to clinical backgrounds but with considerable selection biases. The reviewer is questioned whether a meta-analysis containing such non-randomized heterogeneous studies leads to a reliable conclusion. The overall low incidence of thromboembolic events (1.1%) in the off-OAC patients after successful ablation is an important result to be known, however, a comparison to other cohort should be carefully done.

Response: We acknowledge these points in the discussion section. There are inherent weaknesses of non-randomized studies. Our study was conducted according to the checklist of Preferred Reporting Items for Systematic Reviews and Meta-Analysis (PRISMA) statement. The quality of each study was evaluated using the Newcastle-Ottawa Scale(NOS). Nine of ten included studies were assessed high quality according to the NOS score (range 7 to 9). In the discussion, the statement that “According to subgroup analysis, it showed the discontinuation of warfarin could reduce the risk of TE compared with long-term treatment in patients after AF ablation. One possible reason may be the higher cardiovascular risk profile of the patients in on-OAC group as compared with off-OAC group (e.g. patients in on-OAC group were older age, with a higher CHADS2/CHA2DS2-VASc score and prevalence of persistent or longstanding persistent AF compared with off-OAC group).” To response these comments, we have reviewed all relevant data of included studies again and found that there were no significant differences between two groups regarding the majority of clinical variables. In addition, multivariable Cox regression model was used to identify risk factors independently associated with outcomes in all included studies. In our meta-analysis, heterogeneity among studies was assessed using I2 statistic. When evident heterogeneity was present, we assessed the influence of a single study on the overall pooled effect by omitting one study in each turn. The relevant statistic data were presented in result section. 

Comment: Similar study results have been already reported previously (ref #24 and Romero J et al: JCE 2019 Aug;30(8):1250-1257.). The latter article should be cited in the present study. The authors should address what is a novelty and difference in comparison to these previous studies.

Response: Thank you for your suggestions. We have re-edited it in the revised manuscript. Proietti et al. conducted their search strategy up to July 31, 2018 and included prospective and retrospective trials. Romero et al. conducted their search strategy up to January 2019 and included 5 trials reporting the risk of thromboembolic using CHA2DS2-VASc. These studies provided evidence-based information regarding the anticoagulation strategy after ablation of AF. After these studies were published, however, sever large-scale prospective studies have been published and involve more confident evidence. Hence, we undertook a meta-analysis of the latest and most convincing evidence to assess the safety of discontinuation of OAC after successful AF ablation. Although the main finding of our meta-analysis was consistent with those reported in a previous study by Proietti et al., the differences between ours and the previous ones should be noted. First, our meta-analysis only selected prospective study and included the latest 3 studies (including the largest scale study), which further reinforces earlier outcomes of previous meta-analysis. Second, we performed subgroup anlyses according to study region (non-Asian vs. Asian), sample size (<800 vs. >800), mean age (≤ 60 years vs. >60 years), OAC strategy (warfarin vs. warfarin/DOACs) and follow-up duration (<3 years vs. >3years). Of all subgroup analyses, we found that there is significant difference in OAC strategy with warfarin but not warfarin/DOACs. It may be related to advantages of DOACs. Third, China-AF study ((ref #19) is one of the largest prospective registry studies (including 4512 patients) of Asian patients with AF, which reported TE rates after AF ablation in the off-OAC group were not significant different from the on-OAC group among high-risk patients (female, CHA2DS2 –VASc ≥3; male, CHA2DS2 –VASc � ≥2). Conversely, a meta-analysis by Romero et al. reported that on-OAC after AF ablation with CHA2DS2 –VASc score ≥ 2 is associated with a significantly decreased risk of TE and a favorable net clinical benefit. This discrepancy may be related to follow-up duration and heterogeneity among study designs.

Comment: The residual thromboembolic risk after AF ablation may vary according to the CHADS2/CHADS-Vasc score. Subgroup analyses with regard to these risk scores should be presented.

Response: We fully agree that the residual thromboembolic risk after AF ablation may vary according to the CHADS2/CHA2DS2-VASc score. In our study, we have tried to conduct subgroup analyses according to the CHADS2/CHA2DS2-VASc score. However, after extracting of all relevant data, we found only 2 studies reported outcomes according to risk stratification. One study reported the thromboembolic and major bleeding events according to the CHA2DS2-VASc score (≤1 and ≥2, respectively), the other study reported outcomes with incidence rates per 100 person-years according to risk of thromboembolism and major bleeding (including low-, intermediate- and high-risk). These studies reported similar finding, however, it cannot be pooled analysis directly. We have contacted the authors by e-mail, however, receiving no reply. Finally, we treated it as a limitation and described it in the manuscript.

Comment: A subgroup analysis with DOAC usage may be informative and give a novelty to the present study

Response: Of overall included studies, the anticoagulation used was predominantly warfarin in all but 4 studies, which include partial patients on DOACs. We have performed a subgroup analysis by anticoagulation strategy (warfarin vs. warfarin/ DOACs) in the result section. The result was showed in Table 2. Regrettably, due to lack of sufficient data, we cannot conduct a subgroup analysis with DOAC. It has been described in the limitation section. DOAC is increasingly being used for anticoagulation in patients with AF. Hence, it is important and meaningful to evaluate the efficacy of DOAC in AF patients following successful ablation in the future.

Minor comments

Comment: The discussion section is too long and should be simplified. The first para. is unnecessary because it was mentioned in the introduction.

Response: According to your suggestion, we re-edited the first paragraph of discussion in the revised version.

Comment: L217: Referring the paper by Dr. Bunch, stroke rate in patients who did not undergo ablation would be 3.5%.

Response: Thank you for your reminder. We have corrected it in the revised version.

Comment: English usage should be revised through the manuscript. A native speaker check is recommended.

Response: We have revised the full manuscript regarding English usage.

Reviewer #2

Comment: Although this is interesting study, there are some problems as listed below.

As authors pointed out in “Limitation”, reviewer is afraid of that relatively short follow-up period and low CHADS2 score could affect the results of this study. Precise examination of thromboembolistic risks of patients after catheter ablation for AF would need data of longer follow-up period and one of relatively high-risk patients, for clinical settings. Moreover, the fact that OAC used in many studies was warfarin would also affect the results in this study. Because large number of patients who receive AF ablation are treated by DOACs in these days, data from warfarin usage could have some discrepancy with real world.

 In clinical settings, various patients (CHADS2 score, comorbidity, age, and so on) exist, and complications from OAC usage or discontinuation, such as cerebral infarction and bleeding, are sometimes fatal. Therefore, reviewer thinks that discontinuation of OACs after AF ablation should be based on not crude or heterogeneous data, but detailed and precise one of each patient.

Response: Thank you for your comments. Of overall included studies, the average duration of follow-up ranged from 1.9 to 5 years. To our knowledge, Gaita’s study is one of the longest post-AF ablation follow-ups reported in the literature on this topic, with a median observation time of 60.5 months. Our meta-analysis assessed the safety of discontinuation of OAC after AF ablation according to all relevant published prospective studies and provided the pooled effects regarding the risk of TE and major bleeding. These results may help physician make clinical decisions on this topic. Therefore, it is to be noted that our results could not represent the very long-term risk of TE and major bleeding events, which is needed to be evaluated in the future studies. We agree that the residual TE risk after AF ablation may vary according to the CHADS2 score. A low CHADS2 score may affect the result of this study. Of all included studies, only 2 studies reported the risk of TE according to the risk stratification. These studies showed similar results, which were described in discussion. Proietti et al. conducted the pooled analysis of previous published studies. The pooled analyses demonstrated no statistically significant difference in the risk of TE between patients on-OAC and those off-OAC, whether their CHADS2 was <2 or ≥ 2. Due to lack of relevant data, we did not conduct subgroup analyses according to CHADS2/CHA2DS2-VASc score in our study.

It is the fact that warfarin was used in the majority of included studies. The specific type of OAC was described in Table 1. Of 10 included studies, warfarin as the unique anticoagulant was used for all patients in six studies, the others enrolled part of patients with the use of DOAC. To assess whether the type of OAC would affect the pooled results, we conducted subgroup analysis according to OAC strategy (warfarin vs. warfarin/DOACs). The results showed that there is significant difference between off-OAC and on-OAC group in subgroup with warfarin. However, this statistical difference was not observed in subgroup with warfarin/DOACs. This discrepancy may be related to the use of DOACs. As we know, DOACs are relatively new agents demonstrating superiority or noninferiority to warfarin in reducing risk of thromboembolic complications with similar or reduced bleeding risk. Over the past decade, DOACs have been prescribed more frequently for patients with nonvalvular AF. Up to date, there is no study compare the efficacy of DOAC in patients following AF ablation. Large-scale and adequately powered trials are warranted to confirm these findings.

DOACs have surpassed warfarin as first-line agent because of improved clinical outcomes, lack of monitoring requirements and expanding indication list. Therefore, cost remains a significant barrier to access, especially in people with low and middle income. Our results would provide a reference guide for the clinician in prescribing warfarin for patients after AF ablation.

Of overall included studies, we enrolled AF patients ranged from low-risk to high-risk of TE. Both overall pooled result and subgroup analyses suggest that it may be safe to discontinue OAC therapy in patients after successful AF ablation. However, further large randomized trials are warranted to confirm this. We agree with reviewer that discontinuation of OACs after AF ablation should be based on detailed characteristics of each patient. In addition, previous published evidence on this topic could be treated as a reference.

---

## [Decision Letter · Decision Letter 1]

17 May 2021

PONE-D-21-00884R1

Discontinuation of oral anticoagulation therapy after successful atrial fibrillation ablation: a systematic review and meta-analysis of prospective studies

PLOS ONE

Dear Dr. Liu,

Thank you for submitting your manuscript to PLOS ONE. After careful consideration, we feel that it has merit but does not fully meet PLOS ONE’s publication criteria as it currently stands. Therefore, we invite you to submit a revised version of the manuscript that addresses the points raised during the review process.

We look forward to receiving your revised manuscript.

Kind regards,

Yoshihiro Fukumoto

Academic Editor

PLOS ONE

Reviewers' comments:

Reviewer's Responses to Questions

**Comments to the Author**

1. If the authors have adequately addressed your comments raised in a previous round of review and you feel that this manuscript is now acceptable for publication, you may indicate that here to bypass the “Comments to the Author” section, enter your conflict of interest statement in the “Confidential to Editor” section, and submit your "Accept" recommendation.

Reviewer #1: All comments have been addressed

Reviewer #2: All comments have been addressed

Reviewer #3: (No Response)

2. Is the manuscript technically sound, and do the data support the conclusions?

Reviewer #1: Yes

Reviewer #2: Yes

Reviewer #3: Yes

3. Has the statistical analysis been performed appropriately and rigorously? 

Reviewer #1: Yes

Reviewer #2: Yes

Reviewer #3: Yes

4. Have the authors made all data underlying the findings in their manuscript fully available?

Reviewer #1: Yes

Reviewer #2: Yes

Reviewer #3: Yes

5. Is the manuscript presented in an intelligible fashion and written in standard English?

Reviewer #1: Yes

Reviewer #2: Yes

Reviewer #3: Yes

6. Review Comments to the Author

Reviewer #1: The authors well responded to the comments by the reviewers. The manuscript has been much improved. Finally, please recheck the grammatical especially in the re-edited parts. There are many grammatical errors or insufficient sentences.

Reviewer #2: (No Response)

Reviewer #3: The present is an interesting paper

Some issues

Abstract: kind of studies included should be included (RCTs?; observational studies)

Methods; authors should why they choose Mantel effect

Methods: it should be added if authors pooled studies with analysis corrected at multivariate analysis or not

Results:meta-regression for chads vasc should be performed

Results: warfarin+DOAC is not clear, please write warfarin or DOAC

7. PLOS authors have the option to publish the peer review history of their article (what does this mean?). If published, this will include your full peer review and any attached files.

Reviewer #1: **Yes: **Yasushi Mukai, MD, PhD

Reviewer #2: No

Reviewer #3: **Yes: **Fabrizio D'Ascenzo

---

## [Author Response · Author response to Decision Letter 1]

5 Jun 2021

Response to reviewers' comments

Reviewer #1

Comment: The authors well responded to the comments by the reviewers. The manuscript has been much improved. Finally, please recheck the grammatical especially in the re-edited parts. There are many grammatical errors or insufficient sentences.

Response: Thanks for your suggestions. We have rechecked the grammatical errors and insufficient sentences in the full text.

Reviewer #3

Comment: The present is an interesting paper.

Some issues:

Abstract: kind of studies included should be included (RCTs?; observational studies)

Methods: authors should why they choose Mantel effect

Methods: it should be added if authors pooled studies with analysis corrected at multivariate analysis or not

Results:meta-regression for chads vasc should be performed

Results: warfarin+DOAC is not clear, please write warfarin or DOAC

Response: Thanks for your suggestions. According to predefined inclusion criteria and search results, only prospective studies were included in our study. In addition, no published RCT has evaluated off-OAC vs. on-OAC in AF patient undergoing successful ablation.

There are four used methods of meta-analysis for dichotomous outcomes. Our statistical software implements two random-effects methods for dichotomous data: a Mantel-Haenszel method and an inverse-variance method. The difference between the two is subtle. In practice, the difference is likely to be trivial (Cochrane handbook for systematic reviews of interventions, version 5.1.0). There is a common situation in Cochrane reviews, where the Mantel-Haenszel method is generally preferable to the inverse variance method. Therefore, we chose Mantel-Haenszel method in our study.

In our meta-analysis, all pooled analyses were calculated according to studies’ raw data. The relevant data of each included study were presented in the full text and supplementary materials.

We agree with your suggestion to conduct a meta-regression for CHA2DS2-VASc, which is important and significant. However, only 2 studies reported outcomes according to risk stratification. One study reported the thromboembolic and major bleeding events according to the CHA2DS2-VASc score (≤1 and ≥2, respectively), the other study reported outcomes with the incidence rates per 100 person-years according to risk of thromboembolism and major bleeding (including low-, intermediate- and high-risk). These studies reported similar finding, however, it cannot be used for meta-regression. We have contacted the authors by e-mail, however, receiving no reply. Finally, we treated it as a limitation and described it in the manuscript.

We have changed “Warfarin+DOAC” to “warfarin or DOAC” in the revised manuscript.

We would like to thank the referee again for taking the time to review our manuscript.

---

## [Decision Letter · Decision Letter 2]

11 Jun 2021

Discontinuation of oral anticoagulation therapy after successful atrial fibrillation ablation: a systematic review and meta-analysis of prospective studies

PONE-D-21-00884R2

Dear Dr. Liu,

We’re pleased to inform you that your manuscript has been judged scientifically suitable for publication and will be formally accepted for publication once it meets all outstanding technical requirements.

Kind regards,

Yoshihiro Fukumoto

Academic Editor

PLOS ONE

Additional Editor Comments (optional):

Reviewers' comments:

Reviewer's Responses to Questions

**Comments to the Author**

1. If the authors have adequately addressed your comments raised in a previous round of review and you feel that this manuscript is now acceptable for publication, you may indicate that here to bypass the “Comments to the Author” section, enter your conflict of interest statement in the “Confidential to Editor” section, and submit your "Accept" recommendation.

Reviewer #3: All comments have been addressed

2. Is the manuscript technically sound, and do the data support the conclusions?

Reviewer #3: (No Response)

3. Has the statistical analysis been performed appropriately and rigorously? 

Reviewer #3: (No Response)

4. Have the authors made all data underlying the findings in their manuscript fully available?

Reviewer #3: (No Response)

5. Is the manuscript presented in an intelligible fashion and written in standard English?

Reviewer #3: (No Response)

6. Review Comments to the Author

Reviewer #3: (No Response)

7. PLOS authors have the option to publish the peer review history of their article (what does this mean?). If published, this will include your full peer review and any attached files.

Reviewer #3: **Yes: **Fabrizio D'Ascenzo

---

## [Editor Report · Acceptance letter]

16 Jun 2021

PONE-D-21-00884R2 

Discontinuation of oral anticoagulation therapy after successful atrial fibrillation ablation: a systematic review and meta-analysis of prospective studies 

Dear Dr. Liu:

I'm pleased to inform you that your manuscript has been deemed suitable for publication in PLOS ONE. Congratulations! Your manuscript is now with our production department. 

Kind regards, 

on behalf of

Dr. Yoshihiro Fukumoto 

Academic Editor

PLOS ONE